# Digital Business Models in the Hospitality Sector: Comparing Hotel Bookings with Yacht Charter Bookings

**Helena Zentner** [1] , **Daniela Gračan** [2] **and Marina Barkiđija Sotošek** [2,*]

1    Management Department, Zenadian, 10000 Zagreb, Croatia
2    Faculty of Tourism and Hospitality Management, University of Rijeka, 51410 Opatija, Croatia
*    Correspondence: marinab@fthm.hr

**Abstract:** Within the fast paced digital transformation of the tourism and hospitality sector, the modalities of booking a tourist accommodation have been radically transformed by the uptake of digital business models and digital platforms. This study examines the underlying mechanisms and key specificities of digital business models for two of the sector's sub-segments—hotel accommodation bookings and yacht charter accommodation bookings. Based on the literature review findings, the case study method was applied in relation to key digital business models relevant for hotel bookings. On the other end, in relation to the yacht charter segment and its digital business models, an empirical research was conducted, encompassing a final sample of 162 yacht charter agencies from 42 countries worldwide. The analyzed digital business models have multiple similarities, while there are specific differences between the two hospitality segments. Even though digital business models are highly important in both segments, they are generally more developed and sophisticated in the hotel segment, which is related to the segment's sheer size, in comparison with the younger and smaller yacht charter segment. The novelty is reflected in shedding more light on the characteristics of digital business models in the fast-developing yacht charter segment, including through an empirical study.

**Keywords:** hospitality digital platforms; yacht charter bookings; hotel bookings; digital business models; digital transformation

## 1. Introduction

The increasing diversity of digital business models and the development of the specific concepts such as the sharing economy have complicated online-based consumer behavior. The results of the research conducted consistently show that in the last two decades, the share of online reservations of tourist accommodation has increased very significantly worldwide, as pointed out by numerous authors [1–4]. Indeed, online channels have become crucial for booking tourist accommodation. Likewise, the sharing economy, as an important concept for tourism and hospitality, has grown substantially: in mid-2019, AngelList [5], a U.S. startup website, listed 1546 sharing economy startups with an average valuation of USD 3 million [6]. According to Botsman and Rogers [7,8], consumers are now willing to participate in the sharing economy economic concept and exchange all kinds of resources they own with strangers, including renting out their homes to tourists.

The field of studying business models has experienced rapid growth over the last two decades [9]. Contributing to this trend has been, in particular, the intense development of digital technologies [10,11] and generally a great dynamism of business change, which has led corporate management towards an increasingly pronounced perception of the importance of the business model as a key to the survival and market success of their companies.

Nowadays, the tourism and hospitality industry is developing rapidly and adopting innovative business models, as evidenced by the number of newly established companies that continue to stimulate the market, including both large market segments with many

players and niche market segments that have not yet been fully covered. These conditions have given rise to the virtual hotel operator (VHO) business model as a new option for consumers and hoteliers in marketing their rooms [8] as well as other novel digital business models in the field.

While extant digital business models for hotel bookings have been researched to a solid extent, their counterparts in other hospitality segments, such as the yacht charter accommodation, have been seldomly studied, and the academic knowledge about them remains modest. With the present paper, we aim to address this research gap and contribute to the relevant literature by exploring the following research questions:

RQ1: What are the key differences between digital business models for booking the hotel accommodation and those for booking the yacht charter accommodation, and why do those differences occur?

RQ2: What are the key characteristics of digital business models in these segments?

RQ3: Are these segments served by the same or different companies, and why?

Through a dedicated exploration of these research questions, we aim to shed more light to the increasingly important phenomenon of digital business models, which have been greatly affecting the tourism and hospitality field, as well as enabling more sustainable business practices compared to the previously prevailing business models, particularly in the context of resources utilization and consumption patterns.

It is our intention to contribute to the bodies of literature in entrepreneurship and strategic management by deepening the theoretical and practical understanding of the key characteristics of digital business models that are prevalent in the analyzed sector.

In addition, the paper seeks to identify and explain the differences between a large mainstream hospitality segment and a niche segment in order to foster a greater understanding of the drivers that impact such specific developments. The objective of the study is thus to deepen the theoretical and practical insights into the underlying digital business models by conducting an exploratory research within the two observed segments.

To achieve this aim, the remainder of the paper contains a literature review covering extant knowledge on both of the segments, which is followed by an explanation of the research methodology and presentation of the relevant findings. Thereafter, the comparison between the two segments is discussed and relevant conclusions provided.

## 2. Digital Business Models and Digital Platforms

Authors Härting et al. [12] point out that a business model is the foundation of any company, as it describes how the company works. The concept of business model itself can be viewed from different perspectives, and accordingly, authors define business models in different ways [9,13,14]. Although there is currently no single universally accepted definition of a business model, a systematic analysis of the available literature reveals that there is broad agreement on the key aspects of the term. For the purposes of this paper, the following authors definition will be used: A business model describes the logic of how an organization creates and delivers value to users and how it generates revenue from it. This definition aligns well with the thinking of most authors, is concise, and gets to the core of the concept of a business model.

Among the available frameworks for analyzing the elements and characteristics of business models, the business model canvas (BMC) stands out, which is elaborated in a widely cited book by authors Osterwalder and Pigneur [15] based on Osterwalder's [16] earlier dissertation, where it was scientifically developed. The business model canvas has been accepted both in academia and in corporate practice as a kind of standard for representing and analyzing business models. This framework's theoretical and practical value has been widely recognized, and it has been adopted by numerous academic works by other authors, such as [17–19] and many more. Therefore, this well-established framework will primarily be used in this paper to analyze the characteristics of business models as well as to compare the digital business models of the analyzed segments.

Digital business models are those that create and deliver value primarily through the use of digital technologies, i.e., information and communication technologies [20]. Digital business models differ significantly from traditional business models [21] because digital technologies enable new value propositions as well as new possible modes of operation for virtually all components of the business model. The authors Remane et al. [22] point to several characteristics that distinguish digital business models from the traditional ones, emphasizing the very low marginal costs of digital business models and the significant exponential effects of networks and platforms, while Härting et al. [12] also note that digital business models enable resource savings, process improvements, and cost reductions as well as a greater flexibility in production and shorter timelines for new product introductions. Brousseau and Penard [23] describe digital business models as a combination of three possible roles of platforms, namely intermediation, assembling, and knowledge management, noting that in the case of online travel agencies, all three roles are usually present.

Weill and Woerner [24] note that a well-developed digital business model challenges many assumptions of traditional business models, particularly the business model's reliance on places (physical stores, branches, etc.) and people (salespeople, consultants, etc.) to achieve customer satisfaction. In addition, digital business models have significantly changed business processes, roles in the organization, and the use of data, which are becoming a key resource at the enterprise-wide level. In terms of sustainability, such process optimizations contribute to a more rational use of resources while simultaneously reshaping some of the consumption patterns.

A digital platform, according to Shaughnessy [25], represents a commercial digital network that enables interactions between buyers, bidders, and other participants. The same author notes that such business models, if creatively developed and successfully implemented, can lead to market disruption and an almost unattainable competitive advantage for the company. However, he also notes that there are many misconceptions and false conclusions about platforms in the broader business world. Muzellec et al. [26] point out that the foundation of such Internet businesses is a value proposition for customers on the one hand and a value proposition for service providers on the other. Thus, it is essentially a two-sided market where the key is to balance and attract both sides of the market, i.e., both groups of users. These two groups then offer each other added value through the network effect: the more end customers there are on a particular platform, the more attractive it is to service providers, and conversely, the more providers there are on the platform, the more attractive it is to end customers.

When considering digital platforms, it is important to note that they can be two-sided as well as multisided, meaning that they can mediate between two or more different groups. Accordingly, the company that creates a digital platform very often builds a corresponding "business ecosystem" around it, systematically involving different partners and participants [26]. In the literature, this business is often referred to as a focal enterprise [27] or, more precisely, as the ecosystem leader [24]. The role of the ecosystem leader is to realize the value proposition for all parties in the network as well as to manage its position at the center of these complex relationships [26]. Numerous authors [25,28] point out that the concept of ecosystem is crucial when considering digital platforms and that most successful digital platforms today would be hard to imagine without a developed ecosystem.

## 2.1. Digital Platforms in Tourism and Hospitality

Tourism is one of the first sectors in the process of digital transformation, as its characteristics are very well-suited for the application of digital technologies. Namely, the tourism product is intangible and cannot be tried before purchase [29], and there is often a geographical distance between the customer and the product, and there is an extensive set of information that can be useful for customers in selecting, planning, purchasing, and using a tourism product [30]. These characteristics make the tourism product very suitable

for digitization and open up space for numerous added values that can be realized through the application of modern digital technologies.

Many authors [31–33] point out that information and communication technologies have had a significant impact on tourism, changing the structure of the entire tourism system and opening up a range of new opportunities and threats for all stakeholders. Beynon et al. [34] defined digital tourism as "digital support for the tourism experience before, during, and after the tourism activity". In the context of digital tourism, there are different business models, and Rainhold et al. [35] state that the concept of business model is now central to understanding the way tourism is conducted and to considering possible further changes in tourism.

The application areas of digital business models in tourism include several stages of the tourism experience. Digital business models were first used for the preparation phase of a tourist trip and, in particular, for the digital insight into relevant information and for the reservation of various tourist services and then, over time, extended to the consumption phase of the tourist services themselves and the post-tourist experience phase [36]. Figure 1 shows the three phases of the tourist experience mentioned above, and each of them highlights the fundamental focus of digital business models aimed at this phase.

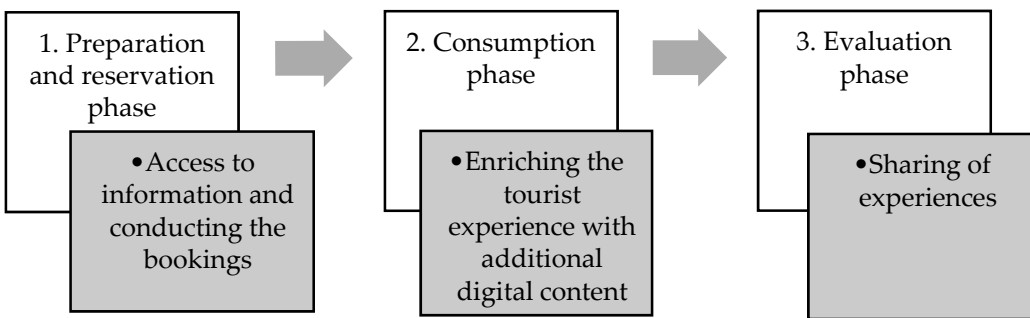

**Figure 1.** Digital business models by stages of tourist experience. Source: Authors' work, stages adapted from Beynon et al. [36].

As can be seen from the figure, digital business models covering the first phase of the tourism experience are primarily focused on providing convenient and easy access to information about the destination, accommodation, transportation, attractions, and other tourism services, i.e., making it as easy as possible to select and book services and prepare for the trip. In the context of this phase, Jensen and Wagner [37] point out that it is an emotionally intense process that requires a significant level of trust due to the intangible nature of services and the need to share personal and financial data when booking. In this first phase, authors Marić and Zoroja [38] point out, among other things, the important role of specific web services but also social networks, which have become important both for the selection of a destination and for the selection of accommodation and other services at the destination. The same authors also describe how it is common for tourists to share impressions, photos, and advice regarding tourism services via web services and social networks, which is also important for the second and third phases of digital support for the tourism experience.

During the actual consumption of tourist services, i.e., during the tourist experience, digital support may include digital guides, maps, interaction via social networks, mobile applications, various audio and video devices, kiosks, etc. [36], and the use of augmented reality (AR) and virtual reality (VR) technologies in tourism is also becoming more developed. Of course, there are also digital business models that place great emphasis on the third phase of the tourism experience, i.e., the evaluation and sharing of impressions after the tourism experience. The results of this phase often serve as a feedback loop for the previous phases and are particularly useful for other tourists when planning trips.

However, considering the segment of tourist accommodation bookings as the narrower object of study in this paper, it becomes clear that the greatest role in this context is played

by digital business models that actively focus on the preparation phase of the tourist trip. Authors Karlsson and Le [39] note that in this sense, a new type of business has emerged, namely online travel agencies (abbreviation OTA), which allow tourists to search for information about various tourist offers and book tourist accommodations and other services online. At the same time, Dutta et al. [4] point out that in the past, airline ticket reservations via OTAs were very frequent, but due to decreasing margins and other changes in the airline market, the activity of most agencies has been mostly focused on tourist accommodation rentals for several years.

The literature shows [37,40] that the process of decision making for booking tourist accommodation is complex, so successful digital service providers for booking tourist accommodations use quite sophisticated approaches in their business models. Virtual communities also play an important role in the decision-making process of tourists [32], where participants share information and advice among themselves, and TripAdvisor is highlighted as the most influential place where such virtual communities exist. Indeed, large amounts of user-generated content (UGC) are created in virtual communities, including user reviews of accommodation establishments, which are an extremely important element for the business of accommodation service providers [32]. Analogously, UGC is very important not only for the accommodation segment but also for other tourism services and is one of the essential features of most successful digital business models in tourism.

One of the frequently encountered characteristics of business models in tourism, especially digital business models, is coopetition. This term refers to a market situation in which there is simultaneous cooperation and competition between the same market participants [41]. Indeed, in the tourism business, there are often situations in which individual companies have mutual and conflicting interests at the same time, which leads to the need to find a balance cooperation and competition, creating a kind of coopetition paradox. Coopetition in tourism has also been studied by [1], who pointed out the existence of horizontal coopetition between similar enterprises and vertical coopetition between different types of enterprises in the tourism value chain. For example, hotel companies cooperate intensively with large booking platforms where they offer their capacities, while at the same time, these hotel companies try to compete directly in the digital market and use various marketing tools to encourage users to book accommodation directly on hotel websites.

In addition to the above-presented overview of extant literature dealing with digital business models in the tourism and hospitality sector in general, the objective of which was to understand the overall phenomena and role of digital business models in the field, the next sub-sections of the literature review will focus more specifically on the two analyzed segments—the hotel segment in Section 2.1.1. and the yacht charter segment in Section 2.1.2. The objective of those sub-sections is to present the extant knowledge that has formed a baseline for designing this research and addressing the research questions.

### 2.1.1. The Uptake of OTA

Authors Krelja Kurelović et al. [3] highlight in particular the role of intermediaries such as Booking.com, Expedia, etc., which "enable hotels to globally distribute their own tourist offer, ensure constant availability (24/7) of the service, flexibly manage prices and reservations in a virtual environment, and interact with visitors in a way that promotes sales". Online travel agencies (OTAs) are thus essentially travel agencies that offer their services exclusively through digital channels [42]. Today, two OTA groups dominate globally [43], each of which includes several well-known market brands. These are:

1. Booking holdings: Booking.com, Priceline, Agoda, Kayak, etc.;
2. Expedia Group: Expedia, Hotels.com, Trivago, Orbitz, Travelocity, etc.

From a business model perspective, it should be noted that not all OTAs have the same digital business model. Indeed, important differences can be identified between "standard OTAs" that use a value chain business model and "two-sided OTAs"—also known as two-sided platforms or markets. Although these differences between the two groups will

be discussed in more detail later with a case study of their best-known representatives, Expedia and Booking.com, they will be briefly explained here. In brief, a standard OTA rents capacity, such as hotel rooms, at wholesale prices and then sells it to end customers through digital channels [44] either independently or as a part of travel packages, much as traditional travel agencies did through traditional distribution channels. In terms of revenue sources, standard OTAs mostly use a "merchant model" where revenue is based on margin, but alternatively, they may use an "agency model" where revenue is based on commissions [43]. In contrast, two-sided platforms typically do not use the merchant model because two-sided platforms do not rent capacity but primarily match supply and demand in their digital market [41,45]. In current business practice, the two-sided platform model has proven to be superior, enabling its leading representatives to grow rapidly and gain a high market share.

Regardless of the business model used, the fact is that OTAs have taken over a large portion of the hotel booking market in recent decades. Analyzing the reasons for the meteoric rise in popularity of online travel agencies compared to traditional agencies, Henne [42] points out that OTAs are able to provide users with interactive options for combining different components of the tourism experience, allowing tourists to virtually create their own personalized arrangement. In addition, the same author points out other benefits of online agencies, such as lower costs and time savings for users. These are just some of the advantages that have opened the space for the accelerated penetration of OTA in the market.

### 2.1.2. Specifics of Yacht Charter Accommodation

The activity of yacht charters includes the rental of vessels for recreational purposes, which mainly involves vessels with accommodation service. When analyzing the charters with accommodation service, which are the focus of this paper, authors [46] divide charter vessels into motor yachts and sailing yachts (sailboats), while in the practice of charter activities, further specific categories of charter vessels have also been distinguished—such as catamarans and gulets.

It is important to point out that charter users rent the whole yacht, not just a room or a suite. At the same time, the charter yacht has a dual function: it is both a housing unit and a means of transportation. As it is a special means of transportation, the aspect of maintenance quality and safety is very important for users because insufficient maintenance of the vessel could endanger users' lives and cause accidents, especially in difficult weather conditions at sea. Considering the risks described above and an increased service variability, it is clear that yacht charter is fundamentally different from conventional accommodation capacity on land.

Moreover, with yacht charter, the period of non-use is much more expensive than with hotel rooms, and a charter yacht is an asset that has an incomparably shorter useful life compared to real estate such as hotels. Therefore, due to the perishability effect in yacht charter segment, there is increased pressure on charter operators to achieve the best possible vessel utilization, and failures in this regard cannot be easily compensated. In addition, the perishability considerations call for more frequent changes in charter fleets, with new yachts periodically being added and old ones removed.

Operationally, yacht charter activity is carried out by specialized charter companies. From the work of Gračan et al. [47], it is clear that a charter company is essentially a company that carries out the activity of renting yachts, where it can rent its own yachts or the yachts of others on the basis of a charter management contract. The specificity of charter companies is that they form and maintain fleets of vessels and commercially manage these fleets, including their technical maintenance and commercial use of their capacities.

In addition to the charter companies, which manage the charter capacities, the companies that fill these capacities are also of great importance for the commercial aspect of the functioning of charter activities. Those are usually specialized travel agencies that offer yacht charter bookings to end customers, the so-called charter agents. Indeed, the majority

of charter boat rentals take place through the intermediation of such specialized agencies, a large number of which are online agencies, i.e., agencies with digital business models. As Babić [48] notes, yacht charter agencies help charter companies rent out their yachts, for which they receive a commission, usually between 15% and 20%. As agencies primarily focus their business on booking charter yachts and systematically attracting end users, digital business models in the charter business mainly occur in charter agencies and are therefore analyzed in the empirical part of the research in this paper.

The importance of digital services in the yacht charter segment is evidenced by previous work [47,49,50] showing that advertising on the Internet is the most important form of advertising for charters in nautical tourism, with a share of over 50%, indicating the importance of digital channels and indirectly digital business models for charter activities in nautical tourism. In Kalči's [50] study, most participants indicated that they prefer to book a yacht charter over the Internet rather than via telephone or face-to-face contact. However, according to the authors' findings based on the results of a detailed literature review, there is insufficient scientific research on digital business models for yacht charters. While recent work has been published on certain aspects of these digital business models [51,52], the aforementioned work does not cover the aspects that are the focus of this paper. Despite the lack of literature, the fact is that today, there is a considerable online activity and a large number of companies pursuing digital business models in nautical tourism, especially in the charter segment. These phenomena are further explored in the present research in order to address the extant research gap and, in particular, answer the research questions.

## 3. Methodology and Approach

In researching this topic, a twofold approach was taken: the case study method for hotel bookings and the empirical study through primary data collection for yacht charter bookings. Firstly, based on the literature review, the case study method was applied in relation to the main digital business models relevant to hotel bookings. The companies selected for the case studies in this paper, Expedia and Booking.com, have somewhat different digital business models, but each is recognized as a leader when it comes to the type of business model applied. A number of authors such as Sjekavica et al. [40], Marić and Zoroja [38], and Krelja Kurelović et al. [3] highlight these companies as the best-known and most represented for online booking of hotel accommodation. It should be noted that the same authors have also identified Airbnb as a highly relevant platform for booking tourist accommodation; however, this platform primarily focuses on homes and apartments and does not enlist hotels, which is why it was not selected for detailed analysis in this study.

In order to conduct the case studies, the authors thoroughly examined the existing works dealing with the businesses in question as well as other available digital sources, including the websites of the observed companies, their mobile applications and social media activities, and the websites of relevant competitors. Scientific methods of synthesis and analysis, induction and deduction, comparison, and classification were applied in processing the data thus collected. For the analysis of the digital business models of the observed companies, the relevant theoretical frameworks for business models were used, primarily the business model canvas as the most widely represented theoretical framework. In this paper, only the most important features of the analyzed business models are emphasized, while the less important features were omitted in order to present the essence of each business model as clearly as possible and allow for an easier comparison between the analyzed companies. Such a concise presentation is in line with the original approach of the authors of the business model canvas, Osterwalder and Pigneur [15].

On the other hand, regarding the yacht charter segment and its digital business models, an empirical research was conducted that included a final sample of 162 yacht charter agencies from 42 countries worldwide. The empirical research in this paper was based on primary data collection, mainly using a questionnaire addressed to the target companies. The questionnaire was written in English, and the data collection was conducted online. During the design of the questionnaire, a phase of preparatory research was also

conducted in the form of pilot research, in which a small number of companies completed the questionnaire, and their feedback was analyzed to improve the content and clarity of the questions before they were sent to all other respondents.

After a series of preparations, reviews, and analysis, data collection on the international population of 932 companies—yacht charter agencies—began in February 2020. It should be noted that the entire available population was invited to participate in the study via electronic means of communication, and several professional reminders were sent out to all companies that still did not submit their answers. The average response rate to the initial invitation to participate in the survey was approximately 3%, and further response rates were similar for each subsequent reminder. A total of up to six reminders were sent to the companies in order to achieve the target sample size and an overall response rate of 18%. A total of 172 completed questionnaires were collected during the research, of which 162 questionnaires were included in the final sample after the initial review, while the remaining 10 questionnaires had to be disregarded due to methodologically relevant issues. Based on the conducted primary research and the obtained answers from the 162 companies, i.e., from nearly 18% of the available population, a business model of a typical yacht charter agency was represented using the business model canvas framework [15].

Finally, the business model canvas was used for comparing digital business models prevailing in the two analyzed hospitality segments as a well-established theoretical framework for representation and analysis of business models that can enable an effective comparison between various business models and their respective elements.

## 4. Findings

In this section the key findings of the research have been presented, starting with case studies' findings for the hotel segment and proceeding towards the empirical findings for the yacht charter segment.

### 4.1. Expedia Platform Case Study

Expedia was founded in 1996, in the pioneering era of online business development, and very quickly distinguished itself as one of the leading online travel agencies. Today, Expedia Group is one of the largest OTA groups in the world and comprises over 200 travel booking websites, including a number of well-known brands such as Hotels.com, Trivago, Travelocity, Orbitz, Vrbo, Homeaway, Egencia, and others [53]. Moreover, some of these market brands are so strong that in many markets, they are even more prevalent than the Expedia umbrella brand, which is evident from the research conducted by various authors [2,38].

The Expedia Group's services include not only accommodation reservations but also flights, cars, and various other tourism services, targeting both individual and business users. Users can create their own travel arrangements by combining different available options, and as Henne [42] points out, there are also services such as a "personalised travel guide" that interactively generates suggested itineraries. Expedia therefore positions itself as a full-service online travel agency selling travel packages as well as flights, hotels, car rentals, cruises, activities, attractions, and related services [53] through a range of brands covering a truly broad spectrum of tourism services.

As a prominent example of a standard OTA, Expedia generates most of its revenue through the merchant model, which means it rents capacity from service providers at wholesale prices and sells it to end customers at a margin either independently or as part of a package deal. Although Expedia also uses a commission-based revenue model in some markets [43], this business model contributes only a small portion of its total revenue, while the majority of revenue is still generated by the company through the merchant model. Figure 2. shows the primary business model of the Expedia platform, using the business model canvas as a framework to illustrate its key features.

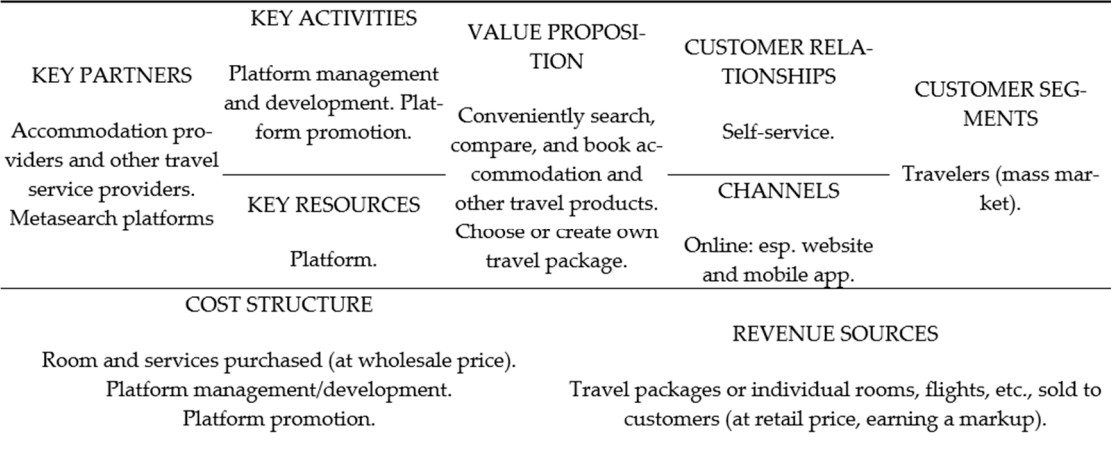

**Figure 2.** Expedia business model. Source: Zentner and Spremić [54].

From the presentation of the business model, it is clear that it is based on a linear value chain in which on the input side are suppliers, providers of tourist services, from whom Expedia purchases capacity at favorable wholesale prices, while on the output side are tourists who buy tourist products or packages from Expedia at retail prices. In this sense, Expedia's business model, although digital due to the online communication and sales channels used and the strong technological infrastructure, is relatively similar to the traditional business models of tourism intermediaries. Indeed, over the years, it has become apparent that it is precisely this linear value chain that has limited Expedia's expansion compared to its main competitor, Booking.com, which has opted for a more scalable two-sided platform business model.

### 4.2. Case Study of the Booking.com Platform

Booking.com was founded in 1996 and is now the world's leading intermediary for hotel accommodation rentals [55], with daily reservations of over one million nights [56]. Although the company has expanded its services to include flight bookings, car rental bookings, and tour and activity bookings, accommodation bookings still account for the majority of its business.

It is well-known that the business model used by Booking.com belongs to the two-sided platform (2SP) category. From the work of authors Tekin Bilbil [41] and Caccinelli and Toledano [45], it can be clearly deduced that the business models of two-sided platforms (2SP) in tourism are significantly different from the business models of standard OTAs. Namely, Booking.com does not actually sell the tourist's accommodation but establishes contact between the tourist and the hotel and mediates the transaction between the two parties. This essentially positions Booking.com as a digital marketplace, not a retailer. The digital marketplace position is a key feature of Booking.com's business model and usually also a key feature of 2SP business models in general.

As a digital marketplace for tourist accommodations, Booking.com must simultaneously appeal to two groups of users: tourists and accommodation providers. These two groups are essentially parties in a two-sided platform where one group's numbers provide value to the other. The more tourists visit Booking.com, the more attractive the platform becomes to accommodation providers, and the greater the supply of accommodation on Booking.com, the more tourists will want to use the platform.

The most important value proposition for tourists is the ability to conveniently search for, compare, and book accommodations from a wide selection, while the most important value proposition for accommodation providers is the large market reach [45] and upselling [43]. Another important aspect of Booking.com's value proposition is the reviews of tourists, which include both qualitative descriptions and quantitative ratings of the accommodations where these tourists have actually stayed [55,56]. The same authors have shown

that tourists' reviews on Booking.com have a significant impact on occupancy rates, i.e., the success of the hotel business, which makes them highly relevant for service providers. At the same time, the number of user reviews collected is a clear result of network effects and sets Booking.com apart from the competition.

Regarding the development of the platform itself and the value proposition, Lopez Kaufman et al. [57] point out that the development of all products and services at Booking.com is based on experimentation with multiple possible versions and ideas, and decisions are made based on the results of these experiments. Furthermore, these authors emphasize that experimentation is an integral part of the organizational culture at Booking.com and that the company has developed a dedicated infrastructure for experimentation as well as a central repository for successful and failed experiments. This work is followed up by Tolga Oztan et al. [58], who add that such a development modality is extremely user-centric. Furthermore, Lako [59] explores the importance of customization, i.e., localization of digital content, which depends not only on language but also on cultural factors, and asserts that Booking.com manages this aspect of digital business quite successfully.

Regarding the openness of the platform, authors Yun et al. [60] examined the extent to which Booking.com uses the concept of open innovation and compared it to its competitor Hotels.com from the Expedia Group. When comparing the openness of the system by supplier (hotels), these authors find that Hotels.com has a much more open system, allowing hotels to access the code and create and connect various additional services. In contrast, the openness of the system on the end-user side was slightly higher for Booking.com. In a general comparison between the two services, Yun et al. concluded [60] that Booking.com stood out especially with flexible payment methods for reserved accommodations.

Another important aspect of Booking.com's business model is that the intermediary services are paid only by the accommodation providers [60], while Booking.com's services are free for tourists. Such a model is often found in two-sided platforms where one party subsidizes the other, and it is precisely such a model that is very commonly used in tourism, including in this platform [16,45]. For example, Booking.com charges accommodation providers a booking commission, which means that their service is free until the moment of booking. However, the booking commission is relatively high and can be up to 30% of the total price [41], while it usually ranges between 10% and 20%.

Figure 3 illustrates Booking.com's business model using the business model canvas as a theoretical framework. From this view, one can clearly see the two-sidedness of the platform, visualized by arrows connecting individual features of PV and SK elements. Indeed, from the business model presented, it is clear that each of the two customer groups addresses a specific value proposition, and these two value propositions are extremely interdependent. The value proposition for tourists is largely based on the widest choice of accommodations, while the value proposition for lodging establishments is based on the widest market reach, from which it follows that one value proposition directly supports and reinforces the other.

Some authors also point out shortcomings in Booking.com's business model. For example, Cacinelli and Toledano [45] point out the regulatory problems faced by this business model: Booking.com has been the subject of a number of anticompetitive allegations and proceedings in Europe, and the company has been forced to change some aspects of its business policy, particularly in relation to price parity in its relationship with service providers. In addition, Sjekavica et al. [40] point to insufficient investment by Booking.com in technological improvements and modernization of services due to its longstanding near-monopoly position.

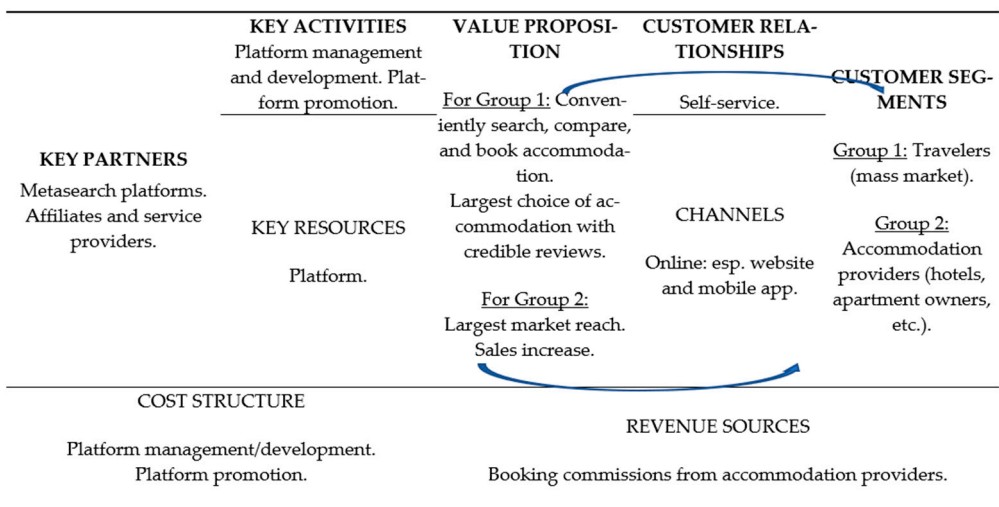

**Figure 3.** Booking.com business model. Source: Zentner and Spremić [54].

*4.3. Empirical Research Findings*

Regarding digital business models in the yacht charter industry, empirical data shows (Table 1) that about 60% of the companies participating in the study predominantly use a digital business model, while the remaining 40% predominantly use a traditional business model.

**Table 1.** Status of digital business model use in the sample.

|  | Number of Companies | Percentage of Companies |
| --- | --- | --- |
| Business model is mostly digital from the beginning of the company | 68 | 42.0 |
| Business model is mostly digital now, but it was not digital in the beginning of the company | 29 | 17.9 |
| Business model is mostly traditional (not digital) | 65 | 40.1 |
| Total | 162 | 100.0 |

Source: Authors' work.

To empirically investigate the key features of digital business models for yacht charters, respondents were asked specific questions on this topic in the questionnaire. The resulting business model canvas for a typical yacht charter agency is shown in Figure 4.

| KEY PARTNERS | KEY ACTIVITIES | VALUE PROPO-SITION | CUSTOMER RE-LATIONSHIPS | CUSTOMER SEG-MENTS |
| --- | --- | --- | --- | --- |
| Charter companies. Central reservation systems. | Digital marketing. Reservation process. Platform maintenance and development | Advice on choosing yachts and destinations-Risk reduction. Large selection of yachts. Practicality. | Self-service. Personal service. | Yachtsmen (most commonly a non-segmented approach within a market niche; alternatively, specialization based on economic or geographic criteria). |
|  | **KEY RESOURCES** The platform. Knowledge and experience. -Customer base. |  | **CHANNELS** Online channels. Direct channels for remote interaction: e-mail, telephone, etc. |  |
| **COST STRUCTURE** Digital marketing. Platform maintenance and development. Human resources | | **REVENUE SOURCES** Commission from yacht bookings. Income from additional services. | | |

**Figure 4.** Business model canvas of a typical online yacht charter. Source: Authors' work.

Regarding the value proposition as the central element of the business model canvas, the following table provides the corresponding responses of the research participants to the question about the important value propositions of yacht charter agencies (Table 2). Based on these results, the four responses with the highest average scores were entered into the typical business model canvas shown in Figure 5.

**Table 2.** Most important values that yacht charter agents provide to clients.

|  | Mean | Std. Deviation | N |
|---|---|---|---|
| Convenience—it is easy to find/select/book a yacht | 4.16 | 0.765 | 160 |
| Accessibility of services over the web, mobile phone, etc. | 3.94 | 0.866 | 160 |
| Cost reduction—lower prices | 3.81 | 0.985 | 158 |
| Great choice of yachts | 4.17 | 0.779 | 160 |
| Advice on selecting yachts and/or destinations | 4.37 | 0.809 | 158 |
| Assistance in organizing travel, flights, etc. | 3.39 | 1.119 | 157 |
| Security and trust—risk reduction | 4.27 | 0.801 | 160 |

Source: Authors' work using the SPSS system.

| KEY PARTNERS | KEY ACTIVITIES | VALUE PROPOSITION | CUSTOMER RELATIONSHIPS | CUSTOMER SEGMENTS |
|---|---|---|---|---|
| **Charter companies.** Central reservation systems. | Digital marketing. **Reservation process.** Platform maintenance and development. | **Advice on choosing yachts and destinations.** Risk reduction. A large selection of yachts. Practicality. | Self-service. **Personal service.** | **Yachtsmen** (most commonly a non-segmented approach within a market niche; alternatively, specialization based on economic or geographic criteria). |
|  | KEY RESOURCES |  | CHANNELS |  |
|  | The platform. **Knowledge and experience.** Customer base. |  | Online channels. **Direct channels for remote interaction: e-mail, telephone, etc.** |  |
| COST STRUCTURE | | | REVENUE SOURCES | |
| Digital marketing. Platform maintenance and development. Human resources. | | | Commissions from boat booking. Income from additional services. | |

**Figure 5.** Business model of a typical online yacht charter agency with marked differentiating features. Source: Authors' work.

The presented table shows that the price component is rated as less important than most others, while the least important component is the organization of travel, flights, etc., which many charter agencies do not offer at all. It is interesting to note that the statement about travel organization has a large standard deviation, suggesting that it is still quite important for some agencies, while for others, it is not important at all. Similarly, the relatively high standard deviation of the cost aspect suggests that agents differ in their assessment of the importance of this value proposition. On the other hand, looking at the categories with the highest mean scores, respondents believe that the most important thing for yacht charter agency clients is their professional advice in choosing a yacht or destination as well as risk mitigation.

It is also evident from the results of the empirical study that charter agencies are mostly specialized in nautical tourism and do not book other types of accommodations or book them only sporadically. The following table (Table 3) shows that more than half of the surveyed charter agencies do not provide tourist accommodation reservation services at all, about 27% of them provide such services only occasionally (they answered sometimes or rarely), and only 9% of them are actively (constantly or often) engaged in tourist accommodation booking.

**Table 3.** Frequency of offering accommodation reservation services in apartments, hotels, etc., by charter agencies.

|  | Number of Companies | Percentage of Companies |
|---|---|---|
| Constantly/Often | 15 | 9.3 |
| Sometimes/Rarely | 44 | 27.2 |
| Not at all | 92 | 56.8 |
| No reply | 11 | 6.8 |
| Total | 162 | 100.0 |

Source: Authors' work.

On the other hand, companies mainly involved in hotel accommodation booking, including the major global platforms analyzed in this study, generally do not offer yacht charter reservation services or provide them rarely and without active advertising. In order to find out the possible reasons why major tourism platforms do not actively offer yacht charter, the questionnaire also asked the respondents a relevant question. The following table (Table 4) shows the respondents' opinions on why major global tourism platforms are not active in the yacht charter segment. As can be seen from the table, the most supported opinion was that the reason is that these companies do not have specific knowledge about the charter market. In addition, many respondents believe that the charter business is too different from the basic offering of these large platforms, and a significant number of respondents also indicated that the charter market appears to be too complicated for large platforms.

**Table 4.** Respondents' opinions on the reasons why large tourism platforms do not offer yacht charter services.

|  | Mean | Std. Deviation | N |
|---|---|---|---|
| Yacht charter market is too small for these large platforms | 2.69 | 1.068 | 161 |
| They think yacht charter market is too complicated | 3.67 | 1.065 | 161 |
| They think yacht charter market is too risky | 3.05 | 1.071 | 161 |
| Yacht charter is too different from the main offering of those platforms | 3.83 | 1.043 | 162 |
| They do not have specific know-how about yacht charter business | 4.01 | 1.143 | 161 |

Source: Authors' work using the SPSS system.

Of course, respondents' opinions on this topic do not necessarily reflect the true reasons for the absence of yacht charters in the offerings of major platforms, but they can be taken as indicative of the views of relevant market participants on this topic. Similarly, these opinions may shed light on the possible reasons for certain differences in digital business models between the two markets observed, as it is clear from the responses presented above that these are two quite different markets and that the pursuit of charter activities requires very specific knowledge. Such an attitude of the respondents is in line with findings from previous literature showing that yacht charter activity is very complex and specific in many respects [61].

## 5. Discussion (Comparing the Segments)

Comparing the characteristics of digital business models of a typical yacht charter agency with the characteristics of typical digital business models for booking hotel accommodation (as presented in Figures 2 and 3), a number of similarities can be identified but also a number of differences. Figure 5 shows the business model canvas of a typical online yacht charter agency, taken from Figure 4, with the addition that the features that differ from those of hotel reservation agencies, that are underlined on the canvas. These differentiating features are further explained below.

The specificity of yacht charter agencies' digital business models is first reflected in their targeted customer segments, which include yachtsmen with all the specifics of this tourist segment. On the supply side, the most important partners are the charter companies. These two differences compared to typical online travel agencies are actually obvious and reflect the essence of yacht charter business.

Another distinctive feature of online yacht charter agencies can be observed within the value proposition field, where, according to the results of empirical research, advice on the selection of boats and destinations is the first priority. Indeed, since renting a yacht is a much more complex and risky decision than renting a hotel accommodation, i.e., since yacht charters are characterized by a substantially higher service variability that cannot be expressed through a simple categorization as in hotels, it is often the case that, in addition to searching for a yacht online, customers also want to receive advice from the agency's professional staff. This characteristic is also related to the other underlined features of the business model, such as the greater role of personal service, which very often complements self-service in online yacht charter agencies, and the significant role of additional communication channels such as e-mail, telephone, etc. These features are very much present in online yacht charter agencies because of the complexity of the product, which often requires a personal consultation so that in the charter business, a large proportion of transactions are still not handled entirely through self-service tools but with additional individual support from professional staff.

This is also reflected in the specific elements of key activities and key resources. Within the key activities, the role of the booking process is quite emphasized, which is a specific combination of personal service and self-service and is the core of the operational business of yacht charter agencies. The role of knowledge and experience is also highlighted in the key resources because yacht charter operations are so complex that they require very specific staff skills that are not commonly available.

Despite the differences highlighted earlier, the digital business model of a typical yacht charter agency also shares many similarities with digital business models for booking hotel accommodation. For example, digital marketing and platform maintenance and development dominate the main activities and cost structure, and the platform is an important resource for both groups of companies. Revenue streams are also quite similar, mostly based on commissions from the completed bookings, with the exception of those travel agencies using the classic merchant model. Finally, there are similarities in the value proposition, where both groups of businesses often offer benefits such as convenience, a wide range of rental options, risk mitigation, etc., although, of course, the rental objects differ significantly.

Overall, it can be concluded that many features of the digital business model of a typical charter agency are relatively similar to the features of online travel agencies, but nevertheless, those are different business models with several distinct features that result from the specificity of the yacht charter product. It is precisely these characteristics of the charter product that have so far prevented effective convergence of these hospitality activities; i.e., they have kept traditional online travel agencies out of the yacht charter segment, and yacht charter agencies have continued to function quite separately from the rest of the tourism market.

## 6. Conclusions

Having structurally analyzed and compared the relevant characteristics of digital business models of the observed segments, it was found that those models have multiple similarities, while there are also specific differences between the two hospitality segments, as detailed in the paper. From the research, it appears that the aforementioned major tourism platforms do not actively handle the yacht charter segment, and yachts for rent may only be found sporadically on these platforms. The possible reasons for this situation are highlighted in the discussion of the results and primarily relate to the specificity of

yacht charter operations and the professional expertise required to successfully handle this activity.

In addition, the research has also shown that multisided platforms have not yet taken root in the charter industry but that digital business models based on the value chain continue to dominate the yacht charter industry. This is an important conclusion in terms of the maturity of digital business models because, as mentioned earlier, multisided platforms generally represent more mature digital business models than linear value chains. Therefore, it can be assumed that the further development of digital business models for the charter industry will move towards multisided platforms, as they are generally a more mature type of business model.

The paper identifies several presumed reasons why multisided platforms have not yet gained traction in yacht charter despite various attempts by individual market participants to establish themselves with such business models. These reasons are mainly due to a niche market and a very specific product, suggesting that the existing business models of the major tourism platforms cannot be easily replicated. Nevertheless, multisided platforms have a chance to take a stronger market share in the yacht charter sector if a genuine, modified digital business model is found that adequately addresses the specifics of yacht charter and if such a business model is fueled with adequate financing to establish its position on the market.

However, it should be noted that this research has several limitations. Firstly, the research focuses on the dominant digital business models, while alternative or emerging digital business models have not been specifically analyzed herein. This represents a further research opportunity that may unveil additional layers of the analyzed phenomena. Secondly, when analyzing the empirical results obtained, it should be noted that these are the perceptions of yacht charter agencies, which do not necessarily fully reflect the attitudes and actions of the customers themselves. Therefore, it is recommended that future research additionally tests the value proposition element and other key questions with customers as respondents to confirm the findings from this side as well. It would also be useful to extend the research to further tourism and hospitality sub-segments in order to obtain a more holistic and complete overview of the entire sector in the context of digital business models. Another interesting research direction would be to track the development of the respective digital business models in time as well as to connect those developments to various internal and external factors that may impact the characteristics and performance of those business models.

Apart from the theoretical contributions of the present research to the relevant bodies of literature by addressing the extant research gap, this paper also provides a practical contribution to the corresponding market participants by enabling them to benchmark their own digital business models and identify potential future course of action for their organizations.

**Author Contributions:** Conceptualization, H.Z. and M.B.S.; methodology, H.Z.; software, H.Z.; validation, D.G., H.Z. and M.B.S.; formal analysis, M.B.S.; investigation, H.Z.; resources, H.Z.; writing—original draft preparation, M.B.S.; writing—review and editing, H.Z. and M.B.S.; visualization, M.B.S.; supervision, D.G.; project administration, D.G. All authors have read and agreed to the published version of the manuscript.

**Funding:** This research received no external funding.

**Institutional Review Board Statement:** Not applicable.

**Informed Consent Statement:** Not applicable.

**Data Availability Statement:** Not applicable.

**Acknowledgments:** The research for this paper was supported by the project Line ZIP UNIRI of the University of Rijeka for the project ZIP-UNIRI-116-2-21 "Influence of exogenous changes on the impact of nautical tourism" supported by the Faculty of Tourism and Hospitality Management, University of Rijeka.

**Conflicts of Interest:** The authors declare no conflict of interest.

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
