# Peer review of "Digital Business Models in the Hospitality Sector: Comparing Hotel Bookings with Yacht Charter Bookings"

_sustainability, doi:10.3390/su141912755_

Round 1

Reviewer 1 Report

Hello,

Interesting, well-written academic paper.  

Good luck!

Author Response

Thank you very much for evaluating and supporting our work, as well as for recognizing it as an interesting and well written paper.

Reviewer 2 Report

The article investigates the effects of different digital business models in terms of differences between hotel bookings and yacht bookings.

SWOT

Thank you for letting me have the opportunity to review your article. Comparing the effectiveness (or applicability) of different business models in different business sectors is in itself an interesting issue. However, the study contains currently several major flaws, in terms of research question development, theoretical development, methodology, and analysis, that need to be addressed in depth to make the study suitable for publication in an academic journal.

Major issues

Introduction

It is not clear what exactly the real (theoretical, academic) issue under scrutiny is (i.e., the definition of the problem currently remains superficial, solely empirical), and why it deserves studying (identification of a knowledge gap, a gap in our understanding, justification of addressing the issue). The fact that something has not been studied before is in itself an insufficient reason to study it. The current research questions appear to be very 'descriptive', i.e., 'what' (and therefore not very helpful in terms of the analysis) questions. Only in the last question (RQ 3) there is a 'why'.

Also, it remains unclear why this study should be published in 'Sustainability', i.e., what exactly is the relationship of the article with sustainability (I assume that it has to do with the 'sharing economy' but this is not clarified)?

As a result of the above, it is also not clear what the (academic (theoretical), practical, societal) contributions of the study are. My suggestion would be to more clearly identify the academic domain (i.e., the domain to which the study intends to make a contribution) of the study (application of digital business models?) clearly identify and justify the gaps studied based on an analysis of the existing literature, i.e., show that and why the gaps need to be investigated. It appears that the tourism sector is 'only' the context within which the phenomenon is studied. This sector appears a worthy case to study the phenomenon.

So, please discuss explicitly what the research question is, how it addresses a gap in our understanding (of the applicability or application of digital business models?) and in which way the article intends to make a contribution to the literature. In other words, I suggest the authors try to embed the study better in the existing literature.

Furthermore, it is important to keep the same frame of reference (level of analysis, perspective, theoretical approach) throughout the article. That implies that it is important to make choices, in terms of your perspective, level of analysis, and theoretical models.

Literature Review

The literature review currently presents the basics of business model theory and digital business models. This presentation is fine, but it remains very broad, and does not, currently, support the study. The presentation does not connect to the research questions, and does not develop a specific theoretical framework that can be used to investigate the research questions empirically. What exactly is or are the objectives that the authors try to achieve in the literature review? Please make these objectives explicit, and show how the literature review is necessary to address the issues, formulate testable hypotheses, or a model that can be investigated empirically.

Methodology

The methodology is presented as a case study method, but it does not seem to be a case study at all. The authors apply the concept 'case studies' in the sense of 'educational case studies'. This is not an academic approach. Cases can, of course, be used as illustrations, but the presented studies do not warrant an academic approach. It is also not clear, why the two examples are chosen, and how they differ. Generally speaking, it is not clearly explained how the chosen methodology will allow the authors to answer the research questions, and how this will lead to valid knowledge development.

It is not clear how the presentation of the empirical data (from the questionnaire) will be leading to the acceptance or rejection of hypotheses, or answer research questions. Only data seems to have been collected from Yacht companies, so how do the authors want to compare this data to 'the hotel' sector (are they comparing the analysis of Expedia and Booking.com with the data collected through the questionnaires?)? These data do not seem to allow such a comparison.

Discussion

The authors seem to make many assumptions regarding the two categories they compare. This makes it very difficult to develop a theoretical understanding of the differences between the business models used in the two segments. They do not succeed, at this stage, in convincing the reader that the insights they present are more than a gist.

Limitations

The limitations of the methodological approach in terms of the academic contribution of the article have not been clarified. How is the validity of the developed knowledge limited, and why? What should we do to 'broaden' the validity, i.e., to make the knowledge applicable to other domains, regions, etc.?

Minor Issues

The authors use a comma rather than a point as a decimal separator in the tables. Please correct.

The authors do not always use an article (the) where needed. Please check. E.g., in the title there should be an article preceding 'hospitality sector'. Also, in the article an article (the) should precede 'hospitality and tourism sector', 'sharing economy', etc.

Author Response

Thank you very much for evaluating our manuscript, as well as for providing further suggestions and comments. Please find below the answers to those comments, section by section:

INTRODUCTION

The introductory section has been extended to address those suggestions more clearly in the revised manuscript.

Among other changes, the importance of studying digital business models in this context has been further explained, the relevant contributions of the study have been additionally articulated – including the bodies of literature to be addressed, the research questions have been further explained and connected to the relevant literature, the sustainability perspective has been added, the explanation how the paper contributes to new insights has been extended etc.

LITERATURE REVIEW

The literature review section has been amended to address those suggestions.

In particular, the objectives of the section have been articulated and the connection of the literature review to the research questions has been emphasized.

We would also like to clarify that in this study we did not formulate specific hypotheses, i.e. the study design was not aiming to test prior hypotheses but rather to conduct an exploratory research of digital business models in the observed business sectors.

Yet, as suggested, in the revised manuscript the literature review section has been more clearly connected to the research questions and the relevant objectives have been articulated.

METHODOLOGY

The manuscript has been further amended to address those suggestions with more clarity.

Regarding the presented case studies, they have been based on prior literature and the analysis of the secondary data. The approach has been explained in the first two paragraphs of the methodology section.

The particular two companies have been chosen because of their clear leadership position in hotel bookings, which was recognized by a number of researchers (as cited in the manuscript – in the 1st paragraph of the methodology section).

The comparison of the two segments was performed by comparing their Business Model Canvases, i.e. the Business Model Canvas framework (as a well established framework for business models) has been used for effectively comparing the business model elements and features of the two segments. This has been specifically explained in the revised manuscript – towards the end of the methodology section. 

We sincerely hope that the above presented arguments, along with the further strengthened explanation in the revised manuscript, will suffice to address these questions concerning the study design.

DISCUSSION

The discussion has been amended to address the sectoral comparison with more clarity.

In addition, we would like to clarify that the mentioned assumptions have been based on relevant prior research as well as the findings from the conducted research.

The comparison of the two analyzed hospitality segments has been based on comparing the relevant Business Model Canvases, as explained in the methodology section.

LIMITATIONS

The explanation of the limitations has been further extended to address those suggestions in the revised manuscript.

In addition, the revised manuscript also contains an indication of the potential further research directions.

MINOR ISSUES

These suggestions have been implemented in the revised manuscript.

Reviewer 3 Report

This is an excellently prepared paper - very well written and presented. I wish the findings were more coherently presented rather than separating the case study and the 'empirical' findings. Also I am intrigued by the word 'empirical' in the section 3.3, as I believe even the case study sections 3.1 and 3.2  were empirical as well.

Well done and all the best.

Author Response

Thank you very much for evaluating and supporting our work, as well as recognizing it as an interesting and well written paper.

With regards to the question related to the word “empirical”, we have used it for the part of our research which was directly based on primary data collection (via survey questionnaire), as opposed to the other part that was based on secondary data analysis. This has now been more clearly explained in the revised manuscript.

Round 2

Reviewer 2 Report

Dear Authors,

Thank you for giving me the opportunity to review the revision of your article. I have provided a range of structured recommendations to help you develop the article further.

Summary

The article intends to compare (the implementation of) digital business models between two touristic sectors. Because no theoretical framework is used for this comparison, and the data from the two sectors are rather incommensurable, the article currently fails to make significant contributions in this sense.

SWOT

The article still appears to address an interesting and timely issue, the variance in (digital) business models among different touristic sectors or sub-sectors. In their revision, the authors have improved and amended the article in a number of commendable ways, but they fundamentally did not address the underlying issues I identified in the first review. As stated in the first review, a very substantial rewrite is needed to make the article suitable for publication in an academic journal.

As indicated in the first review, the problems I have identified are conceptual (theoretical) and methodological (data types, data analysis). As a consequence, as long as these underlying issues have not been resolved the contribution of the article will remain very limited in the three following dimensions; in a societal sense (how does the article contribute to sustainability goals), but also in an academic (how does the article contribute to theory development) or practical (how does the article contribute to improving management practices) sense. As indicated in the first review, the article in its present state lacks academic rigor in both conceptual development and in the design and reporting of the study.

If the authors want to compare business models, or business model implementations between these two sectors, based on the definition of what the authors mean by a ‘business model’ an inventory of the dimensions in which business models can/could vary needs to be established. In the case of the BMC, the various dimensions of the canvas could be transformed into dimensions, but this will not be easy, as the BMC is not a theoretical model. It would then be expected that data collected to compare the implementations in these two sectors are commensurable (they should have the same format, the same source, the same level of analysis) and are measuring variables in line with these dimensions.

The data collected in the empirical part do not correspond with these dimensions in an obvious way: much better explanations of how the authors see the link between the data they have collected and the adaptations they have made in the BMC would be required.

It is also not transparent in which way the data have been collected: what was the sampling method, what was the sample, how representative is that sample, etc.

The authors refer to the Business Model Canvas as their theoretical framework, but the Business Model Canvas is a rather practical tool that was designed to help companies diagnose their business and identify strategic priorities. As such it is not a solid theoretical framework that can be used to identify types or categories of business models: there is even much literature available that criticizes the BMC for being too much focused on one specific type of business model (the classical supply chain organization) and not being suitable in several other domains (service industries, platform-based industries, education, care, public service, etc.) where different business models prevail. In any case, even if the BMC will be used, it will be necessary to provide some further justification for using this framework.

So, major recommendations for improvement are the following:

1.     In order to clarify the contribution of the study:  clearly show what the gap in our understanding (in the literature or in managerial understanding) is, show why this is relevant and to whom, and show how the research questions address this gap. This probably affects the formulation of the research questions and the design of the study. Clearly demonstrate the relevance of the study, i.e., how answering the research questions will lead to a significant contribution in the three domains I mentioned above.

2.     In order to allow a rigorous comparison between the two sub-sectors: build a clear conceptual framework, based on an in-depth review of the literature on (digital?) Business Models, that will help identify different types or categories of business models based on the recognition of a number of dimensions in which business models vary (the research variables).

3.     In order to provide a rigorous methodology: redesign the study in such a way that the obtained data help you in comparing the two sectors on the dimensions identified in the literature review. To obtain valid comparisons, I think it is necessary to collect the same data from both sectors, in the same way. Currently data from one sector are qualitative and from the other sector they are quantitative.

4.     Results could be reported in various ways: the data could be compared in different ways. This depends on the choices you would make. In a quantitative study, I would expect a number of t-tests or Anova’s to identify the (hypothesized) differences. In a qualitative (case-) study, the comparison would of course be qualitative.

I hope that the recommendations above help the authors reconsider the purpose and the research design of the study and provide sufficient input for improving the study. Good luck!

Author Response

Thank you very much for evaluating our revised manuscript, as well as for recognizing the improvements made during the revision process.

We have received and analyzed your further suggestions and comments. Please find our answers and clarifications below.

We do believe there has been a misunderstanding with relation to the methodology and the theoretical framework used in our analysis.

The review suggestions mention that we did not use any theoretical framework in our study, which is not the case: we have applied Business Model Canvas as a well-established theoretical framework widely used in comparable academic studies.

We would like to clarify that the Business Model Canvas is not just a “practical tool that was designed to help companies diagnose their business and identify strategic priorities”. Although it is indeed very popular in the business practice, the Business Model Canvas is primarily a theoretical framework that has been rigorously scientifically developed in Osterwalder’s doctoral dissertation in 2004, and has since been used by numerous researchers in their studies, including in many highly ranking journals.

Therefore, we cannot agree with the claim that the Business Model Canvas is “not a solid theoretical framework that can be used to identify types or categories of business models”. On the contrary, our extensive review of relevant literature, including the origins of that framework, indicate that it is a solid, well-tested and highly relevant theoretical framework that can be used for the business model analysis such as ours.

Of course, there are also some critics of that framework, which is normal for any highly popular tool or concept, but there are many more researchers who support and adopt this framework both in their studies and in their academic lectures. Thus, the Business Model Canvas represents the most widely accepted standard for the representation and analysis of business models – in both academia and practice.

Connected to the above, the review proposes that we develop specific “dimensions” upon which to analyze business models. However, the relevant dimensions (i.e. the 9 elements of Business Model Canvas) have already been scientifically developed and tested, and it is those dimensions that we are comparing in our study.

We would like to clarify that the purpose of our study is not to develop a new theoretical framework for business models, but rather to analyze and compare the specific sectoral digital business models - for which purpose we opt to use a well-established theoretical framework.

Also, we need to comment on the review statement that the Business Model Canvas is “too much focused on one specific type of business model (the classical supply chain organization) and not being suitable in several other domains (service industries, platform-based industries… )”. According to our extensive literature research, this does not hold true. The Business Model Canvas is frequently used for service industries and also for platform businesses, which is visible already from the seminal book of its creators (Osterwalder and Pigneur 2010) where the framework has been clearly used to analyze platform businesses and service businesses. Furthermore, several authors have used this specific framework in the context of tourism and hospitality, which also indicates that it is suitable for our purpose.

Having in mind all the aspects explained above, we firmly believe that the chosen framework is suitable for the objectives of our particular study. Also, development of a new framework is out of scope of our present research.

We sincerely hope for your understanding, in the light of the above stated arguments and explanations, why we will not modify the research design of the present study.

Still, in the revised manuscript, we have extended the explanation about the relevance of the chosen framework, as well as provided an additional explanation and details on the survey methodology applied - which was also suggested in the review comments. The remaining comments in the review were mainly suggesting a redesign of the entire study, which currently does not fit our research concept and scope, as explained in the comments above.

Round 3

Reviewer 2 Report

The article is readable, and makes a very small contribution. Scientific rigor remains low. I leave it up to the editor to decide, but I recommend not publishing it. 

Author Response

Thank you very much for evaluating our revised manuscript, as well as for providing us valuable comments and suggestions.

Following those suggestions, we have additionally improved the manuscript. Please find the revised version in the attachment and our answers in text below:

With regards to sustainability context, in the revised manuscript this has been additionally clarified on pages 2 and 3 of the paper.

With regards to the platforms such as Airbnb and Vrbo, it should be noted that those are peer-to-peer (P2P) platforms, where anybody can enlist their home/property and offer it for rental. However, these P2P platforms typically do not enlist hotel accommodation, which is why they are not directly relevant in the context of this study (namely, our study compares hotel bookings to yacht charter bookings).

In addition, P2P platforms are presently very scarce in yacht charter segment, due to the fact that the required yacht charter field operations (including check-in, check-out, maintenance, field support etc.) are incomparably more complex than for home rentals, making it very difficult for the private owners to organize and manage P2P yacht charters.

Due to the fact that the present research aims to compare hotel bookings with yacht charter bookings, the P2P platforms are not highly represented in neither of these segments, which is why they have not been specifically analyzed in the paper.

This has been briefly explained in the revised manuscript.

With regards to the sampling aspects, it should be noted that the final sample in this research corresponds to around 18% of the total available population, which represents a fairly sufficient sample size.

This has been articulated in the revised manuscript.

With regards to the perishability effects in yacht charter segment (compared to the hotel segment), those have been additionally explained in the revised manuscript.

With regards to the variability in the service delivery, it should be noted that the service variability of yacht charters is indeed quite higher than in hotels (and yacht charter service cannot be easily described with a simple X-stars categorization). Those aspects have been additionally explained in the revised manuscript.